# The Diagnostic Value of microRNA Expression Analysis in Detecting Intraductal Papillomas in Patients with Pathological Nipple Discharge

**DOI:** 10.3390/ijms25031812

**Published:** 2024-02-02

**Authors:** Seher Makineli, Menno R. Vriens, Arjen J. Witkamp, Paul J. van Diest, Cathy B. Moelans

**Affiliations:** 1Department of Surgical Oncology, University Medical Center Utrecht, 3584 CX Utrecht, The Netherlands; mvriens@umcutrecht.nl (M.R.V.); a.j.witkamp@umcutrecht.nl (A.J.W.); 2Department of Pathology, University Medical Center Utrecht, 3584 CX Utrecht, The Netherlands; p.j.vandiest@umcutrecht.nl

**Keywords:** miRNA, biomarkers, nipple discharge, ductoscopy, breast cancer

## Abstract

Patients with pathological nipple discharge (PND) often undergo local surgical procedures because standard radiologic imaging fails to identify the underlying cause. MicroRNA (MiRNA) expression analysis of nipple fluid holds potential for distinguishing between breast diseases. This study aimed to compare miRNA expression levels between nipple fluids from patients with PND to identify possible relevant miRNAs that could differentiate between intraductal papillomas and no abnormalities in the breast tissue. Nipple fluid samples from patients with PND without radiological and pathological suspicion for malignancy who underwent a ductoscopy procedure were analyzed. We used univariate and multivariate regression analyses to identify nipple fluid miRNAs differing between pathologically confirmed papillomas and breast tissue without abnormalities. A total of 27 nipple fluid samples from patients with PND were included for miRNA expression analysis. Out of the 22 miRNAs examined, only miR-145-5p was significantly differentially expressed (upregulated) in nipple fluid from patients with an intraductal papilloma compared to patients showing no breast abnormalities (OR 4.76, *p* = 0.046), with a diagnostic accuracy of 92%. miR-145-5p expression in nipple fluid differs for intraductal papillomas and breast tissue without abnormalities and, therefore, has potential as a diagnostic marker to signal presence of papillomas in PND patients. However, further refinement and validation in clinical trials are necessary to establish its clinical applicability.

## 1. Introduction

Nipple discharge is a common symptom, reported in 4.8–7.4% of women presenting with a breast complaint [1]. When nipple discharge is unilateral, spontaneous, and bloody or serous from a single duct, it is defined as pathological nipple discharge (PND) [2]. The underlying causes of PND are diverse and range from benign conditions, such as lactation, duct ectasia, and papilloma, to malignant conditions [3]. 

In patients with PND, evaluation with mammography and breast ultrasound is performed to rule out malignancy. However, when PND is the only complaint, they both have limited sensitivity [4]. Magnetic resonance imaging (MRI) is a sensitive imaging technique for detecting malignancy, but specificity is low in patients with PND with small lesions [5,6]. Therefore, surgical excision is still required to rule out malignancy in patients with PND without radiological and clinical abnormalities. Nevertheless, the malignancy rate after duct excision surgery is only 8%, meaning the majority of the surgical procedures are performed for benign causes [7]. Ductoscopy is a promising diagnostic and interventional tool used in the workup of PND [8,9]. However, it is not widely used. Also, nipple discharge cytology is limited due to low sensitivity [5,10]. Therefore, the identification of biomarkers in nipple discharge to differentiate between benign lesions and no abnormalities could be essential for improving diagnostics. 

In previous studies, biomarkers for breast cancer derived from nipple aspirate fluid (NAF) have been described as a promising tool for detecting the initiation of carcinogenesis [11,12]. NAF can be defined as a physiological fluid in the breast ductal system that does not spontaneously leave the breast and can be acquired via the nipple by a suction device [13]. PND differs from NAF since PND is spontaneous nipple fluid. In the context of breast disease development, the rationale is that both malignant and benign breast diseases are thought to derive from the epithelial lining of the milk ducts of the breast [14,15]. This makes NAF, as biofluid secreted by the intraductal system of the breast, ideal for investigating the first signs of the development of breast diseases [16]. Many biomarker classes have been investigated in nipple fluid. However, nowadays, NAF-based miRNA assessment in the context of early breast cancer detection is an increasingly investigated research topic because of its high stability in biofluids and its reported association with carcinogenesis [17]. MiRNAs play a critical role in gene regulation, controlling many cellular processes, including cell growth, differentiation, proliferation, and apoptosis [18]. Moreover, specific miRNAs can function as either oncogenes or tumor suppressors [19,20]. MiRNA analysis has not only demonstrated its impact on the pathogenesis of various cancer types [21,22,23] but also on a range of non-malignant conditions such as colon polyps, nasal polyps, benign prostatic hyperplasia [24,25,26]. Data concerning miRNA involvement in certain benign conditions are limited and derived from a few studies. There have been no investigations into the role of miRNA analysis in detecting frequently occurring intraductal papillomas in patients with PND. 

The aim of this study was to compare miRNA expression levels in nipple fluid of patients suffering from PND without radiological suspicion of breast cancer in order to identify possible relevant miRNAs that can discriminate intraductal papillomas from breast tissue with no abnormalities. 

## 2. Results

From May 2019 to August 2020, a total of 35 patients underwent a ductoscopy procedure at our clinic. Of these patients, seven were excluded due to no discharge at time of the procedure (20%), and one patient was excluded because of the presence of DCIS in final histological analysis (2.9%). To evaluate miRNA expression levels, a total of 27 women with PND without radiological and histological suspicion of breast cancer that underwent a ductoscopy procedure were included for analysis: 16 patients with intraductal papillomas and 11 patients with normal breast tissue (no abnormalities). None of the patients had any malignancy in the past 10 years in their medical history. Also, no other disease that may be associated with altered expression of the selected miRNAs was reported.

Baseline characteristics of both cohorts are shown in Table 1. The patient population had a mean age of 50 ± 11.7 years in the intraductal papilloma samples group and a mean age of 42 ± 10.4 years in the no abnormalities group (*p* = 0.09). There were also no significant differences between the other baseline patient characteristics in both groups. The volume of obtained nipple fluid ranged from 10 to 40 μL in both groups. In the intraductal papilloma group, 15 patients (94%) had one productive duct and 1 patient (6%) had two productive ducts. In the normal breast tissue group, nine patients (81.8%) had one productive duct, one patient (9.1%) had two productive ducts, and one patient (9.1%) had three productive ducts.

Of the 22 target miRNAs, only miR-145-5p was significantly differentially expressed (upregulation) between NAF samples from patients with an intraductal papilloma and NAF samples from patients with no intraductal abnormalities (*p* = 0.050) (Table 2). Furthermore, in univariate linear regression analysis, the presence of a papilloma predicted a significant upregulation of miR-145-5p (*p* = 0.012), with a mean fold change of 7.563 (fold changes per interrogated miRNA depicted in Figure 1). 

A multivariable logistic regression analysis was performed including the factors age, evaporation, discharge color, and miR-145-5p normalized expression level. Upregulated miR-145-5p was the only factor significantly and independently predicting the presence of intraductal papillomas, with an OR of 4.76 (CI 1.03–20; *p* = 0.046; n = 27). 

The area under the curve of the ROC- curve showed a diagnostic accuracy rate of 0.920 (CI 0.801–1.000, *p* = 0.002) (Appendix A). The delta Ct mean expression levels for the other miRNAs were not significantly different between both groups. 

## 3. Discussion

To identify potentially relevant miRNAs that can discriminate intraductal papillomas from normal breast tissue, expression levels of 22 human mature miRNAs were evaluated in nipple fluid samples collected from PND patients with and without an intraductal papilloma. This study demonstrated that elevated miR-145-5p levels in nipple fluid predicted the presence of an intraductal papilloma. This suggests that miR-145-5p could help distinguish intraductal papillomas from normal breast tissue in patients without clinical and radiological abnormalities suffering from PND. 

In breast cancer, a variety of miRNAs are known to be down- and upregulated and, therefore, have potential as new biological therapeutic agents, targets, or biomarkers for patient-tailored breast cancer treatment [28,29]. However, there are no data about miRNA expression differences in breast-specific liquid biopsies, such as NAF, between patients with intraductal papillomas and patients with no histological breast abnormalities. This study showed an upregulated expression of miR-145-5p in patients with intraductal papillomas compared to patients with no abnormalities. miR-145-5p is encoded by the *MIR145* gene which is located on Chromosome 5 [30]. This miRNA is mainly considered as a tumor suppressor miRNA in diverse types of cancers and reported to be downregulated in breast cancer [31], bladder cancer [32,33], cervical cancer, renal cancer, and gastrointestinal cancers. However, a few studies reported upregulation of this miRNA in esophageal cancer [34], breast cancer [35,36], and lung cancer [37] and one study reported no difference in expression between normal breast tissue and breast cancer tissue [38]. Moreover, miR-145-5p has been shown to affect the pathogenesis of a number of non-malignant conditions such as aplastic anemia (downregulation), asthma (upregulation), cerebral ischemia/reperfusion injury (upregulation), diabetic nephropathy, and rheumatoid arthritis (upregulation) [39]. Interestingly, according to one study, high expression levels of miR-145–5p were associated with clinical features in breast cancer, such as early menarche, HER2 positivity, and poorly differentiated tumors [36]. 

Previous research results on the relationship between miRNAs and the development/progression of breast diseases have been inconsistent. The exact changes in miR-145-5p expression during the normal-benign-malignant sequence of breast cancer are currently unknown. This study suggests that miR-145-5p may impact the pathogenesis of benign breast disorders such as intraductal papilloma, where it is upregulated, differently than the pathogenesis of malignant breast disorders. While miR-145-5p is generally recognized as a tumor suppressor miRNA that is often downregulated in cancer, two studies have observed upregulated levels of miR-145-5p in the NAF of breast cancer patients [35,36]. The authors related this to the differences (in ethnic backgrounds) of the studied populations, highlighting the need to validate the applicability of this miRNA marker in diverse groups. 

A hypothesis that could explain the upregulation of miR-145-5p in PND samples from patients with intraductal papillomas of the breast is its association with the level of macrophage infiltration and polarization towards M2 and tumor-associated macrophages [40,41]. Macrophages are components of the immune infiltrate within the tumor microenvironment, and they can produce diverse phenotypes in different microenvironments, including alternately activated (M2) macrophages [42]. M2-like cells facilitate tissue remodeling and anti-inflammatory processes and are associated with tumor progression [43]. In the adult, non-pregnant, non-lactating breast, fluid is secreted into the ducts, which may contain exfoliated ductal epithelial cells as well as foam cells, lymphocytes, and neutrophils. Foam cells, thought to be of macrophage lineage, are the most abundant cells found within ductal fluid [44,45]. In the case of nipple discharge, foam cells are also prominent in most samples [46,47]. This is because the presence of an intraductal papilloma can incite an inflammatory reaction within the duct, leading to bloody nipple discharge [48]. Therefore, the upregulation of miR-145-5p in PND samples from intraductal papillomas may be correlated with the level of macrophage infiltration, particularly when compared to PND samples from breast tissue without abnormalities. Nevertheless, further research is needed to validate this hypothesis.

In this study, within the outpatient clinic’s patient population, 20% were excluded due to the absence of discharge at the time of the procedure. However, for the majority of the patients (80%), it was possible to obtain nipple fluid at the outpatient clinic. Previous studies involving NAF collection utilized other non-invasive methods, such as vacuum devices, manual palpation, or hand pumps, and reported successful sample collection ranging from 38 to 90% [13,49,50,51]. In our study population, nasal oxytocin administration was performed before nipple fluid collection. Since these patients suffered from spontaneous nipple discharge, no vacuum device was required for nipple fluid collection. However, considering that in 20% of our study population, nipple fluid collection was not possible, incorporating a vacuum device in these cases may enhance the overall sample collection success rate. Although miRNA expression analysis may still not be suitable for all PND patients, it shows potential as a minimally invasive diagnostic tool in patients with intraductal papillomas with spontaneous discharge and may prevent unnecessary duct excision surgery in patients with benign disease. 

### Limitations

To our knowledge, this is the first study to report on miRNA analysis in nipple fluid from patients with PND. Major limitations of this study were the relatively small sample size and small volume of the analyzed nipple fluid [52]. Larger sample volumes would allow a greater number of circulating miRNA to be obtained, useful for more efficient analysis. Furthermore, we selected 22 human mature miRNAs for evaluation selected from previous investigations identifying them as potential biomarkers for breast cancer. Therefore, other potentially relevant miRNAs may thus not have been discovered in this study. Future prospective research should strive for a careful investigation of a broader range of candidate miRNAs using a non-targeted or multi-targeted approach and strive to collect a larger number of patient samples categorized into three groups: breast tissue without abnormalities, intraductal papillomas, and breast cancer (divided into in situ and invasive cancer). This setup will help identify possible relevant miRNAs that might, in combination, tell benign breast diseases apart from malignant breast diseases with high diagnostic accuracy. Ultimately, this would permit research to advance a step closer to risk prediction in patients with PND. Also, validation of the discriminative value of the miRNAs between benign and normal conditions should be analyzed. Furthermore, additional inflammatory and immune markers can be examined to correlate the miRNA findings. 

## 4. Materials and Methods

### 4.1. Study Design and Population

This prospective cohort study included all consecutive female adult patients with PND without radiological and pathological suspicion for malignancy who underwent a ductoscopy procedure between May 2019 and August 2020 at the University Medical Center Utrecht (UMCU). This study was approved by the Institutional Review Board and the UMC Utrecht Biobank Research Ethics Committee (nr 14-373). All participants provided written informed consent. 

The inclusion criteria were adult women (≥18 years old) with unilateral PND and no radiological suspicion for malignancy referred to UMC Utrecht for ductoscopy. PND was defined as persistent, unilateral, bloody, or serous nipple discharge persisting for at least three months during a non-lactational period. Exclusion criteria were any malignancy in the past 5 years and histopathological confirmation of DCIS or invasive cancer in the biopsied tissue during ductoscopy.

A standard diagnostic evaluation was performed on all patients. This included a medical history, physical examination, and recent radiological imaging (mammography, ultrasonography, and/or MRI and/or core needle/vacuum-assisted biopsy and/or cytology of nipple fluid within three months). All examinations were reviewed and reported by specialized radiologists. 

Patients were eligible for a ductoscopy procedure when radiological and pathological findings were negative. Standard clinical variables were collected, including age at presentation, characteristics of the nipple discharge (laterality and color), and physical exam findings (palpable breast mass and productive ducts). In addition, diagnostic methods, findings from any imaging studies performed, and histopathological details were recorded for each case. Specialized breast pathologists assessed the biopsies.

### 4.2. Nipple Fluid Collection and Processing

Intranasal oxytocin administration was performed before nipple fluid collection [13]. Since patients suffered from spontaneous nipple discharge, no vacuum device was needed for nipple fluid collection. The collected fluid was conserved in a buffer solution (RLT buffer (Qiagen, Hilden, Germany) supplemented with 1:100 *v*/*v* β-mercaptoethanol) at −80 °C until required for analysis. Details concerning sample collection success, sample volume, and sample color were registered. 

### 4.3. RNA Isolation, Reverse Transcription, and Pre-Amplification

To study associations between NAF miRNA expression levels and the presence of intraductal papillomas, the expression of 22 human mature miRNAs were evaluated using Taqman Advanced miRNA assays (ThermoFisher Scientific, Waltham, MA, USA, Catalog number A25576) and Taqman Fast Advanced mastermix (ThermoFisher Scientific) on a ViiA7 real-time PCR system (ThermoFisher Scientific): miR-25a-5p, miR-145-5p, miR-148a-3p, miR-151a-5p, miR-153-3p, miR-155-5p, miR-16-5p, miR-181a-5p, miR-18a-5p, miR-19a-3p, miR-205-5p, miR-21-5p, miR-221-3p, miR-222-3p, miR-29c-5p, miR-30b-5p, miR-320a, miR-339-5p, miR-374b-5p, miR-425-5p, miR-92a-3p, and miR-99b-5p. These oncogenic and tumor suppressor microRNAs were selected for evaluation due to previous investigations and our own pilot study identifying them as potential biomarkers for breast cancer. A search in EMBASE was conducted using search terms related to “microRNA”, “breast”, “intraductal papilloma”, and “benign disease” to identify studies about miRNAs predictive for intraductal papillomas in the breast. Reference lists from articles were also examined to determine related publications. However, since there are no known microRNAs in the literature related to intraductal papillomas in nipple fluid, we opted for the well-known oncogenic miRNAs associated with breast cancer that were reliably measurable in nipple fluid, as a history of benign breast disease can be associated with increased risk of subsequent breast cancer. Tumor suppressor miRNAs miR-22 and Let-7 were not selected. 

miR-99b-5p was used as endogenous control miRNA for NAF, as it demonstrated high expression stability in previous miRNA profiling studies [53,54]. 

First, total RNA was extracted from 10 µL of NAF if available, according to the manufacturer’s instructions, using the AllPrep DNA/RNA/miRNA Universal Kit (Qiagen, Hilden, Germany). Non-human synthetic ath-miR159a (with a 5′ phosphate) was spiked in as procedural control by pre-mixing with RLT plus lysis buffer. Total RNA was eluted in 30 µL RNAse-free water. RNA concentrations were determined with the Qubit RNA HS Assay Kit (Invitrogen, Q32852, Waltham, MA, USA) measured by Qubit 3.0 (ThermoFisher Scientific, Waltham, MA, USA) fluorometric quantification. Evaporation was performed on a subset of the samples after RNA isolation in order to enhance the concentration. For reverse transcription, the undiluted total RNA was poly-A tailed. After adaptor ligation and reverse transcription, cDNA was pre-amplified for 19 cycles using the TaqMan Advanced miRNA cDNA Synthesis Kit (ThermoFisher Scientific, Waltham, MA, USA) on a Veriti 96-well thermal cycler (ThermoFisher Scientific, Waltham, MA, USA). The pre-amplification product was subsequently diluted 10× in 0.1× Tris buffer, pH 8.0, and stored at −20 °C until quantitative PCR. The data were transferred to the Thermo Fisher ConnectTM system (Thermo Fisher Cloud) where miRNA-specific thresholds and baselines were set. The delta Ct value was used for PCR data analysis. Delta Ct was calculated as a difference of the Ct value of the target miRNA and the Ct of the reference miRNA (miRNA-99b-5p): ΔCT = Ct(target miRNA) − Ct(miR-99b-5p) [27]. 

### 4.4. Statistical Analysis

No sample size analysis was performed beforehand due to the limited number of ductoscopy procedures conducted at our clinic. Prevalence and means with standard deviation (SD) were calculated to describe the study population. Depending on distributions, comparisons of continuous variables were performed using Student’s *t*-test or the Mann–Whitney *U* test. Chi-square (χ^2^) test was used to compare categorical variables between patients with intraductal papillomas and no abnormalities. Univariate linear regression analysis was performed for all analyzed miRNAs in intraductal papillomas compared to no breast abnormalities, including the variables age, evaporation, and nipple discharge color. Multivariable logistic regression analysis estimating odds ratio (OR) and 95% confidence interval (CI) was performed to identify factors associated with intraductal papillomas in patients with PND. Variables with an estimated *p*-value of <0.05 in the univariable logistic regression were included in the multivariable logistic regression model. Also, age, nipple discharge color, and evaporation were included in the regression model because previous in-house data showed a significant association between these parameters and miRNA expression in breast cancer. The color of discharge was separated into five groups: colorless, white, yellow, beige, and orange/pink/bloody/green/brown. *p*-values ≤ 0.05 were considered to be statistically significant. The diagnostic accuracy of differentially expressed genes was determined by analyzing receiver operating characteristics (ROC curve). SPSS v29.0 was used to analyze the data for this study.

## 5. Conclusions

In conclusion, the results of this study suggest that miR-145-5p levels in nipple fluid can differentiate between intraductal papilloma and breast tissue without abnormalities in PND patients. This shows the potential of miR-145-5p in nipple fluid analysis as a diagnostic tool in the work-up of patients suffering from PND. Adding nipple fluid analysis with miRNA expression profiling to the diagnostic work-up may help patients with no radiological suspicion of disease to a final diagnosis. However, further refinement and validation in clinical trials are necessary to establish its clinical applicability. In the future, microRNA expression profiling holds promise as a useful tool for optimizing the diagnosis and treatment of PND patients.

## Figures and Tables

**Figure 1 ijms-25-01812-f001:**
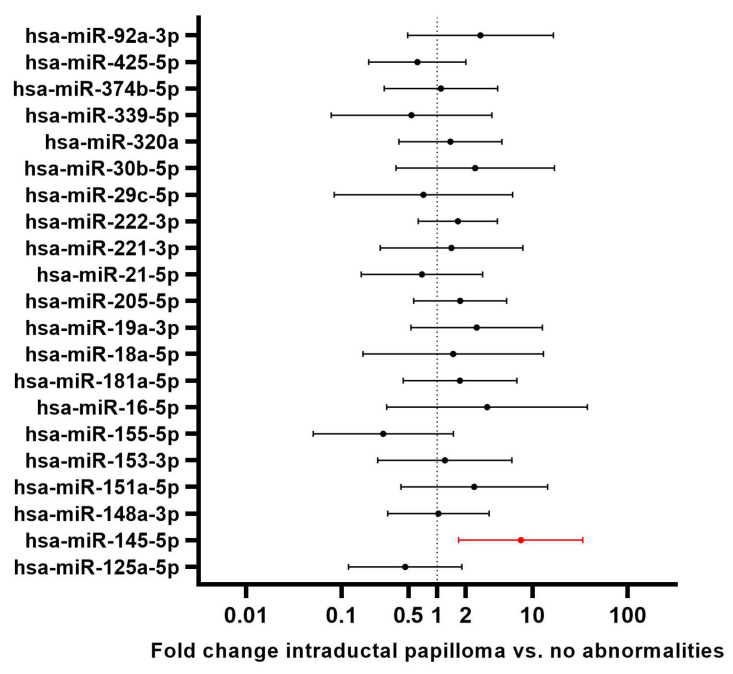
Univariate linear regression-based fold changes and 95% confidence intervals of all analyzed miRNAs in intraductal papillomas compared to no breast abnormalities. MiRNAs with *p*-values < 0.05 were considered of interest for subsequent multivariate analysis. miRNA 145-5p (*p* = 0.012) was upregulated in nipple discharge of patients with intraductal papillomas, with a fold change of 7.56 (CI 1.69–33.90).

**Table 1 ijms-25-01812-t001:** Clinical data and discharge analysis of patients with pathological nipple discharge (PND) undergoing a ductoscopy procedure (n = 27).

Patient Characteristics	Intraductal Papilloma (n = 16)	No Abnormalities (n = 11)	*p*-Value
Age (years), mean ± SD	50 ± 11.7	42 ± 10.4	0.09
Affected breast, left—n (%)	9 (56)	6 (55)	0.93
Volume of collected discharge (µL), mean ± SD	17.2 ± 7.7	15 ± 4.5	0.41
Color of discharge—n (%)			0.30
Colorless	5 (31.3)	3 (27.3)	
White	5 (31.3)	2 (18.2)	
Yellow	3 (18.7)	1 (9.1)	
Orange/pink/bloody/	1 (6.3)	3 (27.3)	
green/brown			
Beige	2 (12.5)	2 (18.2)	
Viscosity—n (%)			0.49
Watery	13 (81.3)	10 (90.9)	
Viscous	3 (18.7)	1 (9.1)	
Cloudiness—n (%)			0.45
Clear	11 (68.7)	9 (81.8)	
Cloudy	5 (31.3)	2 (18.2)	

**Table 2 ijms-25-01812-t002:** Differential miRNA analysis in patients with pathological nipple discharge (PND), subclassified into patients with intraductal papilloma and patients with no breast abnormalities.

miRNA	Intraductal Papilloma (n = 16)	No Abnormalities (n = 11)	*p*-Value
miR-125a-5p			0.264 *
Analyzed (%)	15 (94)	11 (100)	
Mean expression levels (DCT)	−0.446	−1.667	
miR-145-5p			0.050
Analyzed (%)	16 (100)	7 (64)	
Mean expression levels (DCT)	−3.222	−0.779	
miR-148a-3p			0.276
Analyzed (%)	16 (100)	11 (100)	
Mean expression levels (DCT)	−1.188	−2.077	
miR-151a-5p			0.946
Analyzed (%)	12 (75)	7 (64)	
Mean expression levels (DCT)	−1.266	−1.3421	
miR-153-3p			0.941
Analyzed (%)	16 (100)	11 (100)	
Mean expression levels (DCT)	2.512	2.582	
miR-155-5p			0.068
Analyzed (%)	15 (94)	9 (82)	
Mean expression levels (DCT)	1.725	−0.307	
miR-16-5p			0.754
Analyzed (%)	16 (100)	11 (100)	
Mean expression levels (DCT)	−4.354	−3.906	
miR-181a-5p			0.805 *
Analyzed (%)	16 (100)	11 (100)	
Mean expression levels (DCT)	−1.642	−1.740	
miR-18a-5p			0.900
Analyzed (%)	15 (94)	9 (82)	
Mean expression levels (DCT)	0.293	0.135	
miR-19a-3p			0.874 *
Analyzed (%)	16 (100)	10 (91)	
Mean expression levels (DCT)	−2.648	−2.559	
miR-205-5p			0.193
Analyzed (%)	16 (100)	11 (100)	
Mean expression levels (DCT)	−5.780	−4.993	
miR-21-5p			0.204
Analyzed (%)	16 (100)	11 (100)	
Mean expression levels (DCT)	−5.574	−6.673	
miR-221-3p			0.560
Analyzed (%)	16 (100)	10 (91)	
Mean expression levels (DCT)	−3.549	−2.865	
miR-222-3p			0.673 *
Analyzed (%)	16 (100)	10 (91)	
Mean expression levels (DCT)	0.168	0.270	
miR-29c-5p			0.181
Analyzed (%)	14 (88)	8 (73)	
Mean expression levels (DCT)	4.749	3.042	
miR-30b-5p			0.961
Analyzed (%)	13 (81)	11 (100)	
Mean expression levels (DCT)	−3.163	−3.102	
miR-320a			0.402 *
Analyzed (%)	16 (100)	11 (100)	
Mean expression levels (DCT)	−3.3957	−3.608	
miR-339-5p			0.159
Analyzed (%)	14 (88)	9 (82)	
Mean expression levels (DCT)	1.472	−0.346	
miR-374b-5p			0.395
Analyzed (%)	16 (100)	11 (100)	
Mean expression levels (DCT)	0.828	0.117	
miR-425-5p			0.640 *
Analyzed (%)	16 (100)	7 (64)	
Mean expression levels (DCT)	1.380	0.998	
miR-92a-3p			0.961 *
Analyzed (%)	16 (100)	11 (100)	
Mean expression levels (DCT)	−3.864	−3.484	

DCT = delta Ct [27]; * *p*-values based on Mann–Whitney *U* test.

## Data Availability

The datasets generated during and/or analyzed during the current study are not publicly available due to privacy statements but are available from the corresponding author on reasonable request.

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
