# Peer review of "The Diagnostic Value of microRNA Expression Analysis in Detecting Intraductal Papillomas in Patients with Pathological Nipple Discharge"

_ijms, 2024, doi:10.3390/ijms25031812_

Round 1

Reviewer 1 Report

Comments and Suggestions for Authors

This study addressed miRNA expression levels between nipple fluids from patients with pathological nipple discharge to identify possible relevant miRNAs that could differentiate between intraductal papillomas and no abnormalities in the breast tissue.

Compelling evidence have shown that miRNA expression analysis shows potential as a minimally invasive diagnostic and prognostic marker in different types of malignancies as well as various diseases. However, several points in the study design and methodology need to be addressed.

·    I suggest modifying the title, after addressing all the comments to specify what is the potential “value” of the studied miRNA expression analysis, is it diagnostic, discriminative or what?

·    The authors have selected a panel of some oncogenic and some tumor suppressor miRNAs, although a group of breast cancer patients was not included in the study design. This needs clarification.

·    The authors should mention in detail how was the sample size calculated, including the methodology and the references.

·    Other demographic characteristics should be included in the manuscript that are related to breast diseases like family history of breast cancer or any benign breast disease, history of any other malignancy, marital status, parity, history of lactation, the use of hormonal contraceptives, and any other factors related to the development of intraductal papilloma. All these factors should be added and considered in all statistical analysis including logistic regression analysis.

·    History of any other diseases that may be associated with altered expression of the selected miRNAs should be also added and considered in all statistical analysis including logistic regression analysis.

·    The authors need to mention in detail how were the patients differentiated into patoents with intraductal papillomas and those with normal breast tissue.

·    The authors stated that the oncogenic and tumor suppressor microRNAs were selected for evaluation due to previous investigations identifying them as potential biomarkers for breast cancer. However, no studies were cited in the methodology to rationalize the selection of the studied miRNAs.

·    In addition, the selection of the studied miRNAs should not be based only on literature. In-silico analysis through various bioinformatic tools should be done to rationalize the selection of the studied miRNAs.

·    The equation used for the calculation of miRNA expression should be stated with the reference.

·    In Table 2 showing the Differential miRNA analysis in patients with PND subclassified into patients with intraductal papilloma and patients with no breast abnormalities, how is the mean expression levels of some miRNAs has a negative value??

·    The authors reported thar miR-145-5p was significantly differentially expressed (upregulated) between NAF samples from patients with an intraductal papilloma and NAF samples from patients with no intraductal abnormalities with p-value  = 0.050. miR-145-5p is considered as a tumor suppressor miRNA in diverse types of cancers, including bladder cancer, breast cancer, cervical cancer, cholangiocarcinoma, renal cancer, and gastrointestinal cancers. However, few studies have reported up-regulation of this miRNA in some cancers. Moreover, it has been shown to affect pathogenesis of a number of non-malignant conditions such as aplastic anemia, asthma, cerebral ischemia/reperfusion injury, diabetic nephropathy, rheumatoid arthritis and Sjögren syndrome (DOI: 10.1016/j.prp.2022.153780

Accordingly, the patients showing upregulation of this tumor suppressor miRNA should have confirmed exclusion of all other diseases and benign conditions.

·    In addition, more extensive bioinformatics analysis should be performed to highlight possible role of miR-145-5p in the development of intraductal papilloma.

·    The ROC curves should be added to the manuscript, with the sensitivity and specificity and the cut-off value values.

·    In order to be able to validate the discriminative value of the miR-145-5P between benign and normal conditions, a well-established marker known to be elevated in intraductal papilloma should be measured and ROC curves compared.

·    The results should be discussed in more detail, with more references.

Comments on the Quality of English Language

Minor editing of English language required

Reviewer 2 Report

Comments and Suggestions for Authors

This is an interesting work and, I believe, deserving of publication. I have a few comments tht might be considered:

1.) While the choice of miRNA's to test is footnoted it might be of benefit to readersto expand on this in the text, citing the data used to make these selections and their functions.

2.) Were the tissues that were biopsied assessed for miRNA activity? How did that correlate with the fluid results?

3.) Were any inflammatory and/or immune markers examined and did they correlate at all with the miRNA fidings?

Round 2

Reviewer 1 Report

Comments and Suggestions for Authors

The submitted revised version do not show any highlighted modifications relevant to the comments previously sent. Supplemenatry file with ROC curves were not submitted as mentioned in the authors reply.
